# Response of Human Gingival Fibroblasts and *Porphyromonas gingivalis* to UVC-Activated Titanium Surfaces

**DOI:** 10.3390/jfb14030137

**Published:** 2023-02-28

**Authors:** Yin Wen, Hao Dong, Jiating Lin, Xianxian Zhuang, Ruoting Xian, Ping Li, Shaobing Li

**Affiliations:** 1Center of Oral Implantology, Stomatological Hospital, School of Stomatology, Southern Medical University, Guangzhou 510280, China; 2First Clinical Medical College, Xinjiang Medical University, Urumqi 830011, China; 3The First People’s Hospital of Kashgar Region, Kashgar 844000, China

**Keywords:** titanium, dental implants, gingival sealing, biofilm, gingival fibroblasts, *Porphyromonas gingivalis*, ultraviolet, photo-functionalization

## Abstract

Ultraviolet (UV) photofunctionalization has been demonstrated to synergistically improve the osteoblast response and reduce biofilm formation on titanium (Ti) surfaces. However, it remains obscure how photofunctionalization affects soft tissue integration and microbial adhesion on the transmucosal part of a dental implant. This study aimed to investigate the effect of UVC (100–280 nm) pretreatment on the response of human gingival fibroblasts (HGFs) and *Porphyromonas gingivalis* (*P. g.*) to Ti-based implant surfaces. The smooth and anodized nano-engineered Ti-based surfaces were triggered by UVC irradiation, respectively. The results showed that both smooth and nano-surfaces acquired super hydrophilicity without structural alteration after UVC photofunctionalization. UVC-activated smooth surfaces enhanced the adhesion and proliferation of HGFs compared to the untreated smooth ones. Regarding the anodized nano-engineered surfaces, UVC pretreatment weakened the fibroblast attachment but had no adverse effects on proliferation and the related gene expression. Additionally, both Ti-based surfaces could effectively inhibit *P. g*. adhesion after UVC irradiation. Therefore, the UVC photofunctionalization could be more potentially favorable to synergistically improve the fibroblast response and inhibit *P. g.* adhesion on the smooth Ti-based surfaces.

## 1. Introduction

Titanium (Ti) has been the preferred choice for the fabrication of dental implants due to its excellence in biocompatibility, corrosion resistance, and mechanical properties [1,2]. In the Ti dental implant system, a stable transmucosal region of the abutments, necks, or soft tissue integration, plays a critical role in achieving long-term success in dental implant restorations [3,4]. Unfortunately, the peri-implant tissue is susceptible to bacterial invasion compared with the periodontal tissues of natural teeth and the bioinert of Ti materials may aggravate this adverse situation [5,6]. Hence, to achieve better soft tissue integration for Ti implants, efforts should be made to enhance gingival sealing and prevent bacterial aggression.

Various surface modification strategies have been applied to improve soft tissue integration, including machining, acid-etching, argon plasma, laser melting, and anodization [7]. However, most methods are insufficient to inhibit bacterial accumulation because either cells or microbes can be affected by the surface properties of the substrates [8,9]. From the aspect of cells, the anodized nano-Ti surface observed a better adhesion of gingival fibroblasts compared with other types of surfaces. Thereby, the anodization has been considered a promising technique for the surface modification of Ti materials [10,11]. On the other hand, the machined smooth Ti surface displayed less biofilm formation and more convenient plaque control than other surface designs. The surface is characterized as an arithmetic mean roughness value of fewer than 0.2 μm, which is currently recommended for abutments [12,13]. Thus, it would be challenging to fabricate a Ti abutment surface that synergistically improves cell adhesion and inhibits bacterial attachment.

Ultraviolet (UV) photofunctionalization is expected to overcome this issue. It refers to a phenomenon of modification of Ti surfaces that occurs after UV treatment, including the alteration of physicochemical properties and the improvement of bioactivity [14]. Current studies revealed that UV-activated Ti implant surfaces enhanced bone regeneration in vitro and in vivo [15,16]. Meanwhile, UV irradiation, which rarely changes the structure of the objects, has been widely used in surface sterilization for centuries, and it has become attractive for its potential application in light-based antimicrobial therapies [17,18,19]. Interestingly, the effect of UV photofunctionalization may depend on varying factors including the wavelength of UV light, the duration of UV irradiation, the types of materials, the surface properties of the materials, and even the type of organisms. For example, it was reported that Ultraviolet C (UVC; 100–280 nm) pretreatment generated more hydrophilic groups and achieved a higher level of cell adhesion than Ultraviolet A (UVA; 315–400 nm) [20]; the hydrophilic groups increased with a longer duration of UV irradiation [21]; UV pretreatment promoted osteoblastic differentiation of cells on Ti surface with anatase-enriched coating [22]. While many pieces of research have focused on UV irradiation, the effects of UV on Ti with different surface structures, especially the nano-structures, are rarely studied. Moreover, these studies paid little attention to the clinical problems of gingival healing and the accumulation of periodontal pathogens. To the best of our knowledge, it remains unclear how human gingival fibroblasts (HGFs) and *Porphyromonas gingivalis* (*P. g.*) react to the UVC-activated Ti with different surfaces and whether such a response would be beneficial for soft tissue integration.

This study aimed to investigate the effect of UVC photofunctionalization on the biological behaviors of HGFs and *P. g.* on different Ti-based surfaces, which corresponded to two representative types of surface design that are currently applied in commercial abutments. The cellular response of HGFs on different Ti surfaces with related molecular mechanisms concerning the expression of focal adhesion was investigated. Additionally, the response of *P. g.* as well as the co-culture of HGFs with pre-accumulated *P. g.* on different surfaces were also assessed.

## 2. Materials and Methods

### 2.1. Specimen Preparation

Ti discs (commercial grade IV) with a diameter of 15 mm and thickness of 1 mm were used for the following surface treatments. The smooth surfaces were obtained by polishing with multiple sandpapers (Hermes-abrasives, Hamburg, Germany) up to P4000 and silicon rubber burs (0147, Qumo, Wuhan, China), then cleaned ultrasonically (KQ-700DE, Shumei, Kunshan, China) in acetone (GCRF, Guangzhou, China), ethanol (GCRF, Guangzhou, China), and ultrapure water (GOKU-B1, EWLL BIO, Guangzhou, China) sequentially, and then dried at 25 °C, labeled S. The nano-surfaces were obtained by anodization (labeled N), as Wu et al. previously reported [23]. Specifically, the polished and clean samples were immersed in an electrolyte of 0.5% hydrofluoric acid (Macklin, Shanghai, China) for 15 min at 20 V under a direct current power supply system (DP3303D, MESTEK, Shenzhen, China). The UVC pretreatment was performed on the smooth or nano-Ti surfaces using a UVC bactericidal lamp (TUV30W, Philips, Amsterdam, Netherland) with an intensity of 107 μW/cm^2^ (k = 253.7 nm) for 24 h at 25 °C, as Li et al. previously reported [24]. Prepared samples were divided into four groups: (1) S group: smooth Ti surfaces; (2) S + UVC group: smooth Ti surfaces with UVC pretreatment; (3) N group: nano-Ti surfaces; and (4) N + UVC group: nano-Ti surfaces with UVC pretreatment.

### 2.2. Surface Characterization

The physicochemical properties of the samples were characterized, as mentioned previously [25,26]. The surface topography was analyzed by field-emission scanning electron microscopy (FE-SEM; Sigma 300, Zeiss, Oberkochen, Germany). The surface roughness was assessed by atomic force microscopy (AFM; Bruker dimension icon, Bruker, Karlsruhe, Germany). The surface chemistry of the samples was investigated by energy-dispersive X-ray spectroscopy (EDS, XFlash 6|30, Bruker, Karlsruhe, Germany) and X-ray photoelectron spectroscopy (XPS; K-Alpha, Thermo Fisher Scientific, Waltham, MA, USA). Wettability was measured by static water contact angles (WCA) on the sessile-drop method measuring device (DSA XROLL, Betops, Guangzhou, China).

### 2.3. Response of Human Gingival Fibroblasts

#### 2.3.1. Cell Culture and Seeding

Cell cultivation followed the protocols provided by Gibco^®^. HGF (CRL-2014; ATCC, Manassas, VA, USA) was obtained commercially and cultured in Dulbecco’s modified eagle medium (DMEM; Gibco, Waltham, MA, USA), which was supplemented with 10% fetal bovine serum (FBS; Gibco, Waltham, MA, USA) and 1% penicillin–streptomycin (Gibco, Waltham, MA, USA) and incubated at 37 °C, 5% CO_2_. The medium was changed every 2 days. Passages 3–7 were used in the following experiments.

#### 2.3.2. Initial Cell Adhesion and Cell Proliferation

Initial cell adhesion was evaluated by nucleus counting following the protocol provided by Meilun. HGFs were seeded at 1 × 10^4^ cells/cm^2^ on the samples in 24-well plates. After 0.5, 1, and 2 h of incubation, samples were rinsed gently with phosphate-buffered saline (PBS; Gibco, Waltham, MA, USA) twice following the fixation in 4% paraformaldehyde (PFA; Meilun, Dalian, China) for 15 min and staining with 1 μM 4′,6-diamidino-2-phenylindole (DAPI; Meilun, Dalian, China) for 10 min at 25 °C. Cells were observed and documented using an inverted fluorescence microscope (DMIL LED, Leica, Wetzlar, Germany).

The tetrazolium-based assay was used to assess the relative cell proliferation, as followed by Dojindo Laboratories. HGFs were seeded at 2 × 10^4^ cells/cm^2^ on the samples in 24-well plates. After 1, 4, and 7 d of incubation, the samples were washed in PBS twice and added to 600 µL of fresh medium plus 60 µL of CCK-8 solution (Dojindo, Kumamoto, Japan) per well for 3 h of incubation at 37 °C, 5% CO_2_. The acquired fluid samples were collected in a 96-well plate (100 µL per well) and then measured by a microplate reader (SpectraMax Plus384, Molecular Devices, San Jose, CA, USA) at 450 nm. To quantify the relative cell proliferation, the samples without UVC pretreatment were set as the controls.

#### 2.3.3. Cell Spreading with Focal Adhesion

The morphology of cell spreading with the distribution of focal adhesions was visualized by immunofluorescence staining following the experimental protocol provided by Proteintech^®^. HGFs were seeded at 2 × 10^4^ cells/cm^2^ on the samples in 24-well plates. After 2 and 24 h of incubation, cells were fixed with 4% PFA for 15 min, permeabilized with 0.2% Triton X-100 solution for 5 min, blocked with 5% goat serum for 30 min at 37 °C, and then incubated with the anti-focal adhesion kinase (FAK) primary antibody (1:100, Proteintech, Wuhan, China) overnight at 4 °C. After that, samples were incubated with the Alexa Fluor^®^ 488-labeled secondary antibody (1:2000, AAT Bioquest, Pleasanton, CA, USA) and Alexa Fluor^®^ 594-Phalloidin (1:1000, AAT Bioquest, Pleasanton, CA, USA) for 1 h, respectively. Finally, samples were immersed in a mounting medium with DAPI. Images were captured by a confocal laser-scanning microscope (STELLARIS, Leica, Wetzlar, Germany). Biometric analysis was performed based on these images.

#### 2.3.4. Adhesion-Related Gene Expression

The reverse transcription–quantitative polymerase chain reaction (RT-qPCR) assay was performed to investigate the adhesion-related gene expression levels of HGFs, which followed the Minimum Information for Publication of Quantitative Real-Time PCR Experiments (MIQE) guidelines [27]. HGFs were seeded at 2 × 10^5^ cells/cm^2^ on the samples in 24-well plates. After 2 and 24 h of incubation, total RNAs were isolated using Trizol reagent (Accurate Biology, Changsha, China), and cDNAs were generated using the Reverse Transcription Mix Kit (Accurate Biology, Changsha, China). Quantitative polymerase chain reaction (qPCR) was conducted using SYBR Green Premix Kit (Accurate Biology, Changsha, China) on a real-time PCR system (LightCycler 96, Roche, Basel, Switzerland). The relative changes in mRNA expression determined from qPCR experiments were calculated by the 2^−ΔΔCT^ method. Glyceraldehyde-3-phosphate dehydrogenase (GAPDH) was used as the reference gene. The groups without UVC pretreatment were set as the controls. The primers (Tsingke Biotechnology, Beijing, China) used are shown in Table 1.

### 2.4. Response of Porphyromonas gingivalis

#### 2.4.1. Preparation of *Porphyromonas gingivalis*

Microbial culture followed the protocols provided by AOBOX. *P. g.* (33277, ATCC, Manassas, VA, USA) and was maintained in the brain heart infusion (BHI) broth (AOBOX, Beijing, China) that was supplemented with 5 μg/mL hemin (AOBOX, Beijing, China) and 5 μg/mL vitamin K (AOBOX, Beijing, China) and incubated in an anaerobic jar (C-31, MGC, Tokyo, Japan) at 37 °C. After incubation, the *P. g.* suspension was adjusted to an optical density (OD)_600_ = 0.15 by the microplate reader for counting the colony-forming units (CFUs), as Di Giulio et al. previously described [28].

#### 2.4.2. Bacterial Viability and Biofilm Formation

The viability of *P. g.* was evaluated by live/dead bacterial staining following the protocols provided by AAT Bioquest. *P. g.* strains were seeded at 1 × 10^8^ CFU/mL of 2 mL on the samples in 24-well plates for 24 h. After incubation, samples were gently washed with saline (Kelun, Chengdu, China) and stained using a Live/Dead Bacterial Assay Kit (AAT Bioquest, Pleasanton, CA, USA). Images were recorded by the inverted fluorescence microscope at an emission wavelength of 530 nm for MycoLight™ Green and 660 nm for propidium iodide.

Biofilm formation was measured by crystal violet staining, as Kamble et al. previously reported [29]. Specifically, *P. g.* were seeded at 1 × 10^8^ CFU/mL of 2 mL on the samples in 24-well plates for 24 h. After incubation, samples were fixed with 99% methanol for 15 min and submerged in 0.5% crystal violet solution (Macklin, Shanghai, China) for 20 min, following extensive washing with ultrapure water to remove the residual dye. After drying out, the samples were added with 33% acetic acid (600 μL per well) and gently shaken for 30 min to release the bound crystal violet. Finally, the OD values were measured at 590 nm.

### 2.5. Co-Culture of Human Gingival Fibroblasts and Porphyromonas gingivalis

The adhesion of HGFs under *P. g.* pre-accumulation was explored by a co-culture model using the established protocol provided by Gao et al. [30]. *P. g.* suspended in BHI was priorly added onto the samples at 1 × 10^8^ CFU/mL of 2 mL in 24-well plates and incubated in an anaerobic jar at 37 °C for 60 min. Subsequently, HGFs were seeded at 2 × 10^4^ cells/cm^2^ in DMEM supplemented with 10% FBS and 2% BHI without antibiotics at 37 °C, 5% CO_2,_ for 2 h. After incubation, the samples were assessed by fluorescence staining with Alexa Fluor^®^ 594-Phalloidin and DAPI, which was processed as mentioned above.

### 2.6. Statistical Analysis

Each experiment was performed in triplicates. Data were analyzed by SPSS 25. 0 (SPSS, Chicago, IL, USA). Statistical difference was analyzed using the student’s *t* test. The level of *p*-value < 0.05 was considered significant.

## 3. Results

### 3.1. Surface Characterization

The physicochemical properties of all the groups are summarized in Table 2.

The surface roughness of the smooth-based Ti was no more than 20 nm in Sa while the nano-Ti was slightly rougher, which turned out to be 20–30 nm in Sa (Table 2). The smooth-based Ti surfaces exhibited a smooth, flat, and scratch-like pattern with parallel polishing lines in low magnification (2000×) (Figure 1A). Under high magnification (50,000×) (Figure 1C), some irregular pits were distributed among the polishing lines. By contrast, the scratching texture was not observed on the nano-based Ti surfaces in low magnification (Figure 1B). Under high magnification, it displayed a wide range of tubular structures in tight and orderly arrangement (Figure 1D). Each unit of the structures was approximately 90 nm in diameter (Table 2), like empty nano-scale test tubes that opened at the top and closed at the bottom, suggesting the existence of TiO_2_ nanotube arrays. No significant changes in topography or roughness were observed on smooth or nano-Ti surfaces after UVC pretreatment.

Surface wettability was measured by the static WCA based on the established principle: WCA > 90° was described as hydrophobic; WCA < 90° was considered hydrophilic; and WCA that was very close to 0° ascribed to the superhydrophilicity [31]. As shown in Table 2 and Figure 1G and H, the static WCA was 103.20° (±0.93) on the smooth Ti and 47.75° (±0.38) on the nano-Ti. After UVC pretreatment, both smooth and nano-Ti surfaces displayed overspreading of the water droplets with a static WCA at 0°. Therefore, the smooth Ti surfaces were hydrophobic while the nano-Ti surfaces were hydrophilic. After UVC pretreatment, both surfaces acquired superhydrophilicity.

The main chemical compositions of the samples were all related to the O, C, and Ti elements, which were measured by the EDS and XPS analysis (Table 2). EDS showed that both the S + UVC group (Figure 2A) and the N + UVC group (Figure 2B) exhibited an increased content of oxygen and a decreased content of carbon compared with the control groups. This result was confirmed by XPS. As illustrated in the peak fitting images of XPS, the O 1s peak (Figure 3A,B) was resolved into two individual peaks. The major peak at 530.00 eV corresponded to the TiO_2_ phase, and the second peak located at a binding energy of 532.15 eV was attributed to the hydroxyl group. The C 1s peak (Figure 3C,D) at approximately 284.8 eV was ascribed to the carbon contaminants. After UVC pretreatment, both smooth and nano-Ti surfaces displayed a higher peak of a hydroxyl group from O 1s peaks (Figure 3A,B) and lower C 1s peaks (Figure 3C,D) compared with the untreated samples. Therefore, after UVC pretreatment, both surfaces exhibited a higher proportion of hydroxyls and a lower percentage of carbohydrate contaminants compared with the untreated ones.

### 3.2. Response of Human Gingival Fibroblasts

#### 3.2.1. Initial Cell Adhesion and Cell Proliferation

After 0.5, 1, and 2 h of incubation, the S + UVC group (Figure 4A) was found to have a significantly greater number of adhered HGFs compared with the S group (*p* < 0.001 at 0.5 and 1 h; *p* = 0.018 at 2 h) (Figure 4C). In contrast, the N + UV group (Figure 4B) displayed a lower number of adhered HGFs compared with the N group at 0.5, 1, and 2 h, with significant differences at 0.5 h (*p* < 0.001) and 1 h (*p* = 0.005) (Figure 4D).

The relative cell proliferation of HGFs in the S + UVC group (Figure 5A) was significantly higher than the S group for 1 d (*p* = 0.011), 4 d (*p* < 0.001), and 7 d (*p* < 0.001). By contrast, the results showed no significant difference between N + UVC and N groups (Figure 5B) in the relative proliferation rate at 1, 4, and 7 d (*p* > 0.05).

#### 3.2.2. Cell Spreading with Distribution of Focal Adhesion

After adhering to the Ti surfaces, HGFs spread with a rearrangement of the cytoskeleton (in red) and an expression of the focal adhesive structures (in green). Representative fluorescent images for 2 h of incubation time showed that HGFs in the S + UVC group (Figure 6B) extensively stretched out with thick actin filaments and a peripheral protrusion of filopodia. In contrast, the S group (Figure 6A) exhibited HGFs with poorly elongated shapes. Similarly, HGFs in the N + UVC group (Figure 6D) extended better than in the N group (Figure 6C), in which most of the cells still showed a shrunk and non-elongated morphology. A biometric analysis showed that the cell area of HGFs at 2 h (Figure 6I) was significantly larger in the S + UVC group (*p* < 0.001) or N + UVC group (*p* = 0.002) compared with the untreated groups. After 24 h of incubation, HGFs on all the surfaces (Figure 6E–H) were thoroughly elongated in a spindle shape with more protruded filopodia and a higher expression of focal adhesive structures, especially at the leading edge of the protrusion. No significant differences in cell morphology were observed among these groups at 24 h. Therefore, UVC pretreatment promoted the early spreading of HGFs, both on smooth and nano-Ti surfaces.

#### 3.2.3. Adhesion-Related Gene Expression

After 2 h of incubation, the S + UVC group (Figure 7A) displayed a higher mRNA expression of FAK, integrin β_1_, and vinculin compared with the S group, with significant differences in integrin β_1_ (*p* = 0.009) and vinculin (*p* < 0.001). The N + UVC group (Figure 7B) exhibited insignificantly different mRNA levels of FAK and integrin β_1_ compared with the N group, although a higher level of vinculin was observed (*p* = 0.180). After 24 h, the S + UVC group (Figure 7C) showed higher mRNA levels of integrin β_1_ (*p* = 0.133) and vinculin (*p* = 0.105), with a lower mRNA level of FAK (*p* = 0.391) compared with the S group, while the N + UVC group (Figure 7D) exhibited similar mRNA levels of FAK, integrin β_1_, and vinculin (*p* = 0.105). It should be noted that no significant differences were observed in all the groups regarding the mRNA expressions of FAK, integrin β_1_, and vinculin at 24 h (Figure 7C,D).

### 3.3. Response of Porphyromonas gingivalis

The adhered *P. g.* was stained by live/dead assay after 24 h of incubation (Figure 8A). After UVC pretreatment, both smooth Ti and nano-Ti surfaces (Figure 8B) demonstrated a reduction in the live/dead rate of *P. g.* (*p* < 0.05). Furthermore, the biofilm formation was evaluated by crystal violet (Figure 8C). After UVC pretreatment, both smooth Ti (*p* = 0.002) and nano-Ti (*p* = 0.049) surfaces showed that the OD values of crystal violet decreased, indicating that the adhered *P. g* inhibited.

### 3.4. Co-Culture of Human Gingival Fibroblasts and Porphyromonas gingivalis

After 2 h of co-culture, HGFs (cytoskeleton in pink and nuclei in light blue) adhered to both smooth-based and nano-based Ti surfaces, which were previously covered with the *P. g.* colonies (diameter < 5 μm in light blue). The S + UVC group (Figure 9B,F) had more extensive cell spreading and less bacterial accumulation compared with the S group (Figure 9A,E), while the N + UVC group (Figure 9D,H) failed to show a significant change in cell spreading compared with the N group (Figure 9C,G), although the *P. g.* pre-accumulation declined.

## 4. Discussion

UV photofunctionalization has been reported to improve cell adhesion and reduce biofilm formation on Ti-based surfaces, which may potentially be favorable for the soft tissue integration of Ti dental implants. In this study, smooth Ti and nano-Ti surfaces were chosen as two representative surface designs that are currently used in commercial abutments. Compared with the smooth-based Ti surfaces (Figure 1A,C) that exhibited scratch-like patterns on a micro scale, the nano-based Ti surfaces (Figure 1B,D) displayed a wide range of TiO_2_ nanotubes with increased surface roughness (Figure 1E,F). It was previously evidenced that the introduction of nano-texture augmented the contact area and increased the surface wettability, which had a higher affinity for cells [32]. However, such multi-structured topography could be less effective for inhibiting bacterial accumulation and removing bacterial debris than the smooth surface [13]. For this reason, UVC pretreatment was applied in this study to optimize the bioactivity and antimicrobial activity of different Ti surfaces. After UVC pretreatment, both smooth and nano-Ti surfaces achieved a static WCA at 0° (Table 2) as evidence of a super-hydrophilic transition (Figure 1G,H) without significant changes in surface topography or roughness, which was consistent with the previous research [33]. This phenomenon was explained by the removal of carbohydrate contaminants and the production of hydrophilic chemicals [25]. It was also confirmed in our study by the results of EDS (Figure 2) and XPS (Figure 3), which exhibited increased hydroxyl groups (Figure 3A,B) and decreased carbon elements on the UVC-pretreated surfaces compared with the untreated ones. The related mechanism of the alteration in chemical properties may differ in the wavelengths of UV light. It was reported that UVA induced TiO_2_ photocatalytic reactions while UVC generated hydrophilic groups by directly breaking the chemical bonds rather than photocatalysis [34].

HGFs belong to the major cellular constituents of the fibrous connective tissue and are responsible for collagen synthesis, which is essential for soft tissue integration [35]. In this study, UVC pretreatment encouraged the adhesion (Figure 4B), spreading (Figure 6), and proliferation (Figure 5A) of HGFs on smooth Ti surfaces, which broadly supported the work of other studies [20,36,37]. UV-induced superhydrophilicity accelerated the adsorption of extracellular molecules onto the substrates ahead of cell adherence, which contributed to an increased expression of focal adhesions in gingival fibroblasts [38,39,40]. Focal adhesion was considered as the mode cells contacting the implant surfaces [41]. It was initiated by the specific recognition of integrins with the pre-adsorbed extracellular matrix. As a result, FAK is recruited to the sites of integrins for activating phosphorylation to regulate the adhesion, spreading, proliferation, and migration of cells [26]. Additionally, vinculin played an essential role in the stabilization of focal adhesion, which acted as a linker between integrins and the cytoskeleton and was related to mechanical signal transduction [42]. As expected, our results showed that UVC pretreatment upregulated the mRNA levels of FAK, integrin β_1_, and vinculin at 2 h on the smooth Ti surface (Figure 7A), indicating that the UVC-pretreated smooth Ti improved the expression of focal adhesion. Nonetheless, the mRNA level of FAK was slightly lower on the UVC-pretreated smooth Ti compared to the untreated one at 24 h (Figure 7C). It was consistent with the previous findings that the focal adhesion turnover might relate to the activation of cell migration and proliferation [43]. Thus, these results suggested that the UVC pretreatment enhanced the adhesive and proliferated behaviors of HGFs on smooth Ti.

Notably, our results also showed that the UVC-pretreated nano-Ti surface, compared with the untreated one, exhibited a lower speed of initial cell adhesion before 2 h (Figure 4C). Although it showed a similar degree of cell adherence with an increased extent of cell spreading after 2 h, the proliferation activity within a seven-day period (Figure 5B) displayed no significant difference between these two groups. Moreover, the results of RT-qPCR showed insignificant mRNA levels of FAK and integrin β_1_ at 2 h between the UVC-pretreated nano-Ti and untreated ones (Figure 7B), although the change in the vinculin level conformed to the results of cell spreading (Figure 6). These results suggested that UVC pretreatment discouraged the initial adhesion of HGFs on the nano-Ti surface but had no significant adverse impact on cell proliferation. Undoubtedly, it was contradictory to our study that carried on the smooth Ti, and it was also different from the previous research which found that UVC pretreatment improved the initial adhesion of bone marrow cells and endothelial cells on the nano-porous network Ti surfaces [44]. Despite the types of cells, another possible reason might relate to the influence on wettability by surface structures. It has been extensively reported in the literature that surface topographies could severely modify surface wettability [45]. For example, a superhydrophilic surface with hierarchical structures exhibited a lower degree of cell adhesion than a moderate hydrophilic surface [46]. Such a phenomenon was explained by the rapid fluid spreading and hydrated layer formation, which consequently hindered the protein adsorption and cell attachment [32]. Nevertheless, the present study failed to demonstrate a distinction between the surface wettability of smooth and nano-samples by the detection of static WCA (Figure 1G,H). For proving this hypothesis, a dynamic measurement of WCA may be required. Moreover, other explanations such as the switched protein conformation induced by structural modification or UV irradiation should also be noticed [47,48].

*P. g.* is reportedly correlated with the occurrence of peri-implantitis [49]. In this study, UVC pretreatment inactivated *P. g.* both on smooth and nano-Ti surfaces (Figure 8), which is consistent with previous studies [50,51]. Related mechanisms were previously explained by the following facts. One of these was related to the production of reactive oxygen species (ROS) or direct damage to the DNA [37]. Another theory was associated with the formation of the hydrated layer as mentioned above [52]. On the other hand, the present study also demonstrated that under a condition of *P. g.* pre-accumulation, UVC pretreatment still enhanced the initial adhesion and spreading of HGFs on the smooth Ti (Figure 9A,B,E,F). While a reduced number of *P. g.* was observed on the UVC-activated nano-Ti compared with the non-activated nano-Ti, the impact on the early response of HGFs was insignificant (Figure 9C,D,G,H). These results might imply that UVC pretreatment had the potential to enhance the initial adhesion and spreading of HGFs on smooth Ti under the accumulation of *P. g*. On the nano-Ti surfaces, UVC pretreatment might reduce the pre-accumulated *P. g.* as well as maintain a good biocompatibility for HGFs at an early stage. It was confirmed with the recent study showing that UVC photofunctionalization improved the biocompatibility of TiO_2_ surfaces, which were covered by the bacterial remnants. Nevertheless, our study was only undergone by a short-term observation and failed to show further changes in cellular functions in this co-culture model. It remains unknown how long this antibacterial effect lasts in an oral environment. It is also unclear whether the production of ROS would have a positive or negative impact on the peri-implant tissue, even though the tissue was not directly exposed to UVC during the pretreatment. While UV irradiation was proven to be safe and effective in anti-microbial photodynamic therapies by generating ROS [53], it could be different from UVC pretreatment when regarding the timing and dosage of UV. Therefore, the profound effects of UVC pretreatment need to be figured out in future studies. Other issues including time and cost efficiency as well as ozone pollution should also be considered.

In summary, our findings demonstrated that UVC pretreatment had different effects on HGFs while it had a similar tendency on *P. g.* between smooth and nano-Ti. Nevertheless, the biological response to the implant materials was extremely complicated, and it is still poorly understood for the optimal design of dental implant abutments. Our findings can provide new insights into the potential application of UVC pretreatment on Ti surfaces for enhancing soft tissue integration and preventing peri-implant diseases. The major limitations of this study are as follows. Firstly, the mechanism for the inhibited adhesive behavior of HGFs on UVC-pretreated nano-Ti has been not confirmed yet. Secondly, the molecular mechanisms of changes in the function of HGFs should be further explored. Thirdly, it remains unknown whether the long-term effects of UVC pretreatment exhibit the pros and cons of soft tissue sealing because most in vitro models were observed in short periods. In addition, the effect exerted on a flat disc sample might be significantly different from that on the curved abutment surface. More importantly, the present study lacks in vivo verification. To find out more evidence for promoting the clinical translation of UV pretreatment, the questions raised in this study may need to be solved in the future.

## 5. Conclusions

Within the limitations of this study, UVC pretreatment enhanced the adhesion and proliferation of HGFs and reduced the accumulation of *P. g.* on the smooth Ti surface compared with the untreated smooth Ti surface. Although the UVC-activated nano-Ti surface discouraged the initial adhesion of HGFs, it had no adverse effects on the proliferation of HGFs and still displayed an inactivation to *P. g.* compared with the untreated condition. It is suggested that UVC pretreatment could potentially be more favorable for improving the adhesion of HGFs onto smooth Ti. Further studies may be required for confirming the long-term effectiveness and biosafety of UVC pretreatment under the consideration of clinical applications.

## Figures and Tables

**Figure 1 jfb-14-00137-f001:**
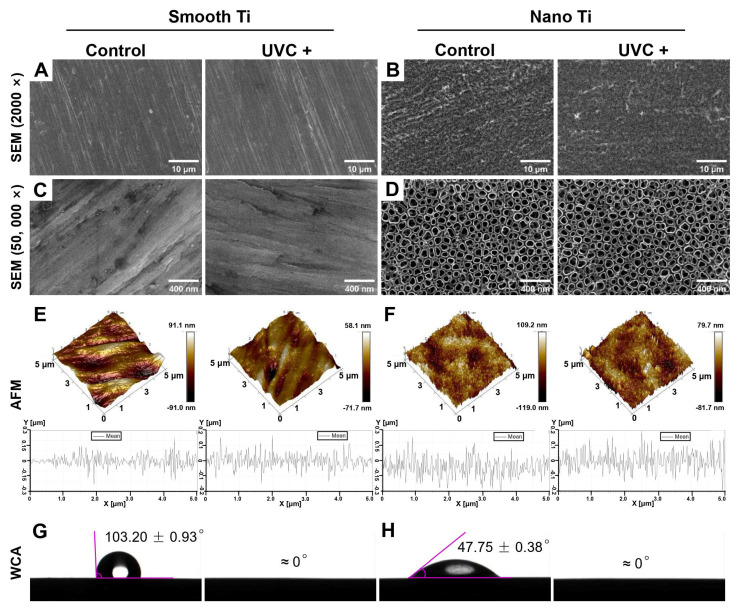
Surface characterization of samples after UVC irradiation: representative SEM images of (**A**) smooth-based Ti (magnification 2000×), (**B**) nano-based Ti (magnification 2000×), (**C**) smooth-based Ti (magnification 50,000×), and (**D**) nano-based Ti (magnification 50,000×); AFM images with the surface roughness profile at a range of 5 μm × 5 μm for (**E**) smooth-based Ti and (**F**) nano-based Ti before and after UVC irradiation; representative images of water contact angle measured on (**G**) smooth-based Ti and (**H**) nano-based Ti, before and after UVC irradiation.

**Figure 2 jfb-14-00137-f002:**
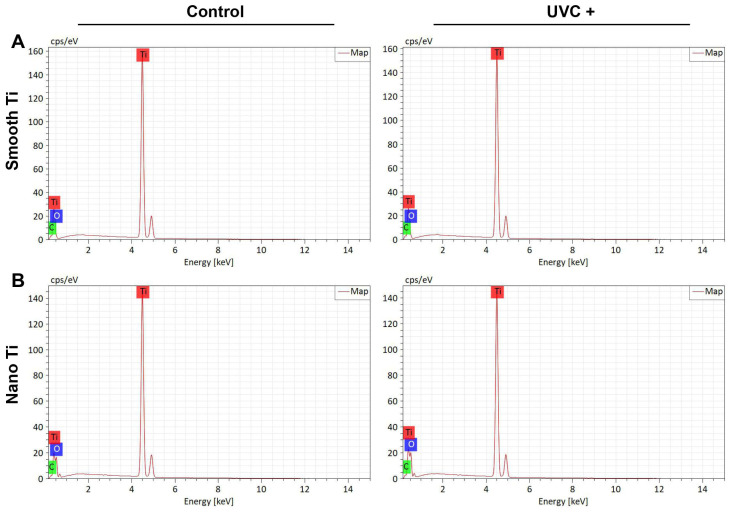
Element peaks of EDS analysis for (**A**) smooth Ti and (**B**) nano-Ti.

**Figure 3 jfb-14-00137-f003:**
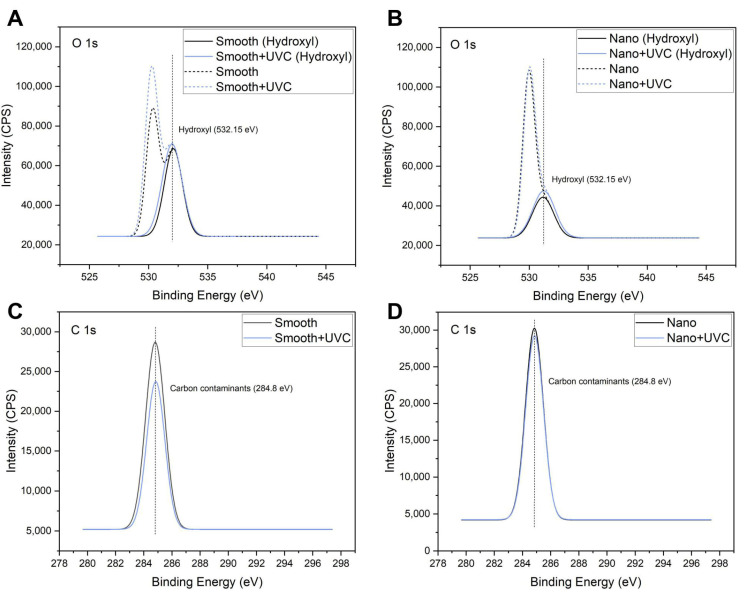
Peak fitting of XPS for the samples: (**A**) O 1s for smooth-based Ti, (**B**) O 1s for nano-based Ti, (**C**) C 1s for smooth-based Ti, and (**D**) C 1s for nano-based Ti.

**Figure 4 jfb-14-00137-f004:**
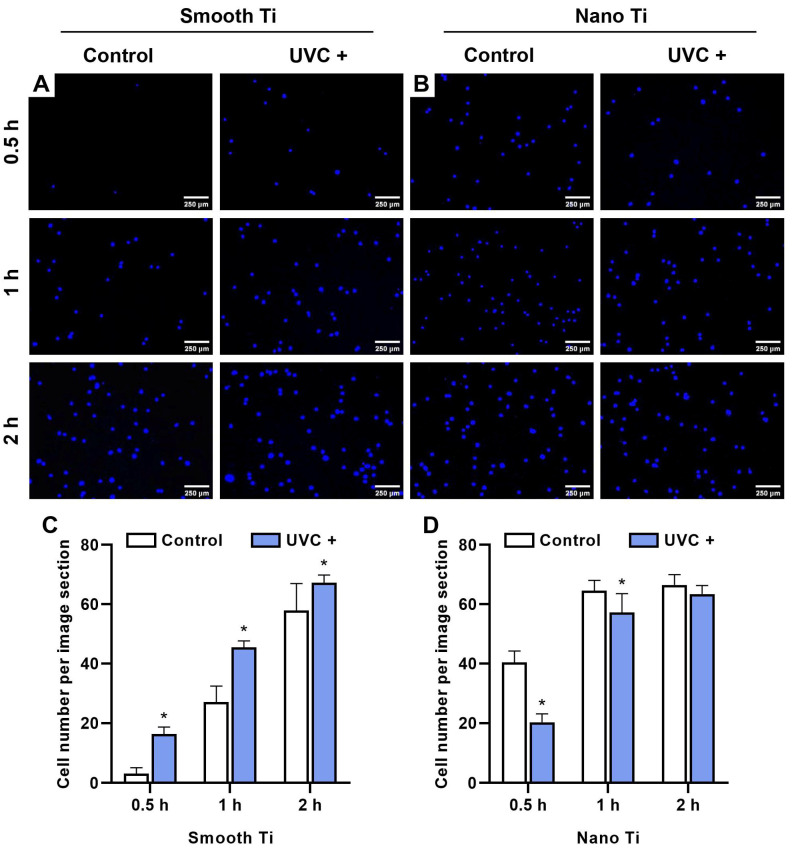
Adhesion of HGFs on the samples. Representative fluorescent images of (**A**) smooth-based Ti and (**B**) nano-based Ti for 0.5, 1, and 2 h, stained by DAPI. Quantitative analysis of adhered cells per image section (2.5 mm^2^) of (**C**) smooth-based Ti and (**D**) nano-based Ti for 0.5, 1, and 2 h. * represents *p* < 0.05 when compared to the control.

**Figure 5 jfb-14-00137-f005:**
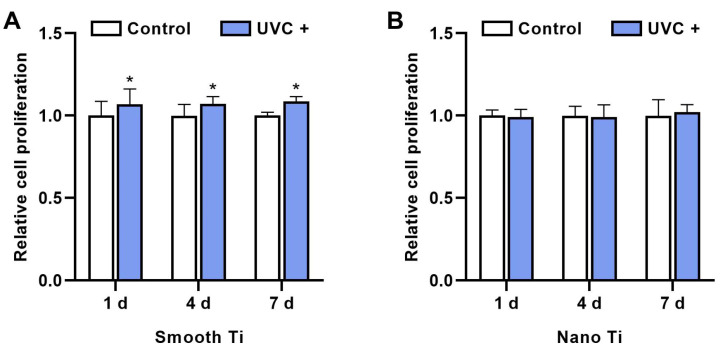
The relative proliferation rate of HGFs on the (**A**) smooth-based and (**B**) nano-based Ti surfaces for 1, 4, and 7 d, determined by CCK-8 assay (λ = 450 nm). * represents *p* < 0.05 when compared to the control.

**Figure 6 jfb-14-00137-f006:**
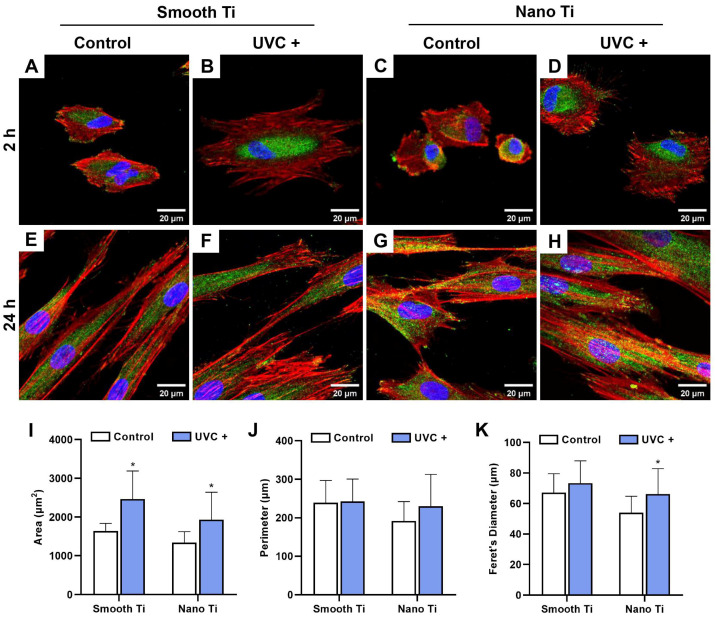
The spreading of HGFs on the samples with immunofluorescent staining for cytoskeleton with phalloidin (red), FAK (green), and nuclei with DAPI (blue): representative confocal microscopic images of the spread of HGFs at 2 h for (**A**) smooth Ti, (**B**) smooth + UVC Ti, (**C**) nano-Ti, and (**D**) nano + UVC Ti, and at 24 h for (**E**) smooth Ti, (**F**) smooth + UVC Ti, (**G**) nano-Ti, and (**H**) nano + UVC Ti; biometric analysis of (**I**) cellular area, (**J**) perimeter, and (**K**) Feret’s diameter at 2 h. * represents *p* < 0.05 when compared to the control.

**Figure 7 jfb-14-00137-f007:**
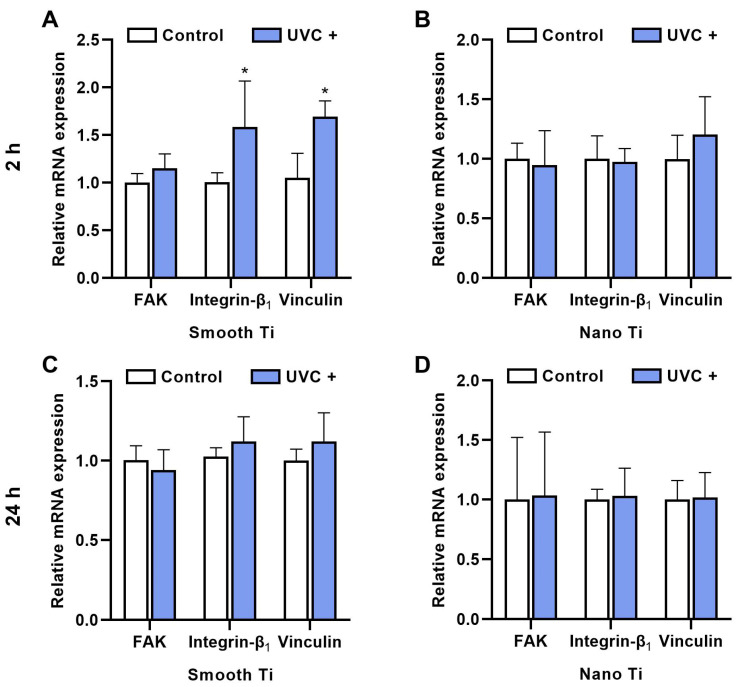
Relative mRNA expression (FAK, integrin β1, and vinculin) of HGFs on the samples of (**A**) smooth-based Ti and (**B**) nano-based Ti for 2 h, (**C**) smooth-based Ti, and (**D**) nano-based Ti for 24 h. * represents *p* < 0.05 when compared to the control.

**Figure 8 jfb-14-00137-f008:**
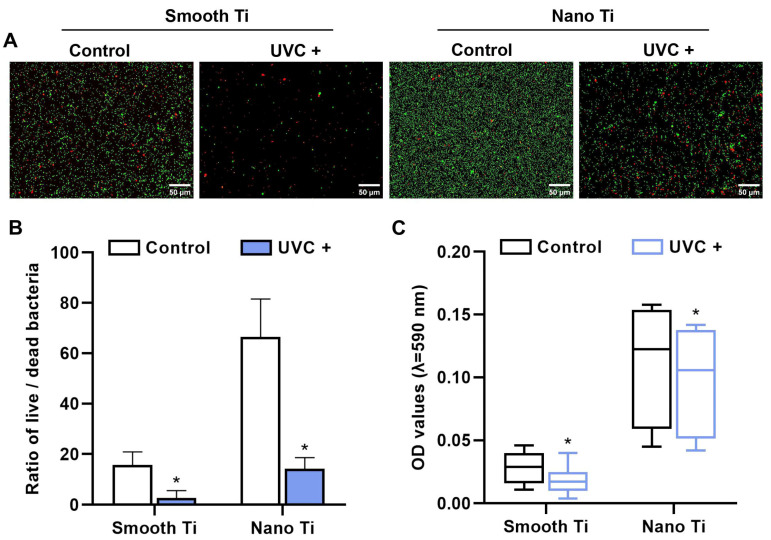
Response of *P. g.* on the sample surfaces. (**A**) Representative images of *P. g.* viability by fluorescent staining for live bacteria (green) and dead bacteria (red). (**B**) Quantitative comparison of live/dead ratio based on (**A**). (**C**) Quantitative comparison of biofilm formation by OD values of crystal violet with the subtraction of a blank. * represents *p* < 0.05 when compared to the control.

**Figure 9 jfb-14-00137-f009:**
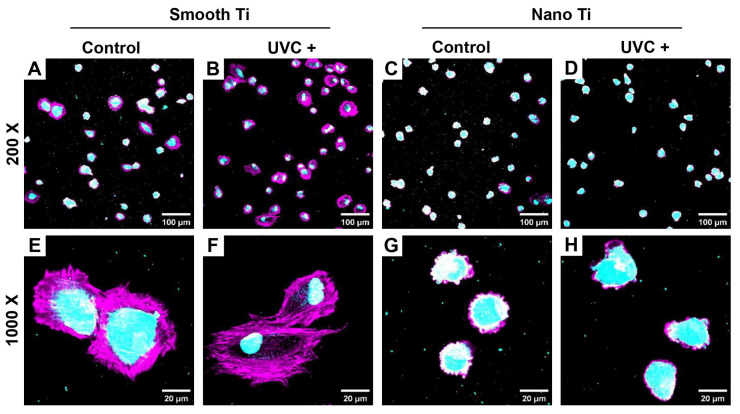
Representative fluorescent images of HGFs co-cultured with pre-accumulated *P. g.* at 2 h by fluorescent staining for cytoskeleton with phalloidin (pink) and dsDNA of cells and bacteria with DAPI (light blue): images in magnification 200× for (**A**) smooth Ti, (**B**) smooth + UVC Ti, (**C**) nano-Ti, (**D**) nano + UVC Ti and in magnification 1000× for (**E**) smooth Ti, (**F**) smooth + UVC Ti, (**G**) nano-Ti, and (**H**) nano + UVC Ti.

**Table 1 jfb-14-00137-t001:** Primers used for a real-time polymerase chain reaction in this study.

Gene	Forward Primer Sequence (5′-3′)	Reverse Primer Sequence (3′-5′)
FAK	GCTTACCTTGACCCCAACTTG	ACGTTCCATACCAGTACCCAG
Integrin β_1_	CCTACTTCTGCACGATGTGATG	CCTTTGCTACGGTTGGTTACATT
Vinculin	CGAATCCCAACCATAAGCAC	CGCACAGTCTCCTTCACAGA
GAPDH	GGAGCGAGATCCCTCCAAAAT	GGCTGTTGTCATACTTCTCATGG

**Table 2 jfb-14-00137-t002:** Physiochemical properties of the samples.

**Physiochemical Properties**		**Groups**			
		**S**	**S + UVC**	**N**	**N + UVC**
Nanotube diameter (nm)		-	-	95.66 ± 9.52	94.52 ± 8.06
Surface roughness (nm)	Sa	13.85 ± 5.05	18.00 ± 6.20	27.10 ± 3.91	23.77 ± 7.02
	Sq	18.07 ± 6.85	23.60 ± 8.47	33.80 ± 4.55	30.13 ± 8.58
EDS atomic percentage (%)	O	16.11 ± 1.12	17.99 ± 0.88	47.99 ± 0.76 ^B^	50.62 ± 0.80 ^b^
	C	4.35 ± 0.26	3.81 ± 0.46	1.45 ± 0.03	1.55 ± 0.06
	Ti	79.54 ± 1.35	78.20 ± 1.10	50.56 ± 0.78 ^B^	47.83 ± 0.85 ^b^
XPS atomic percentage (%)	O 1s	50.94 ± 0.38 ^A^	57.05 ± 1.19 ^a^	44.77 ± 0.25	46.23 ± 0.52
	C 1s	33.35 ± 1.63 ^A^	24.35 ± 0.34 ^a^	34.15 ± 0.99	33.57 ± 1.85
	Ti 2p	16.76 ± 0.79	18.60 ± 1.34	20.09 ± 0.13	20.14 ± 0.36
Water contact angle (°)		103.20 ± 0.93	≈0	47.75 ± 0.38	≈0

Sa: arithmetic mean height; Sq: root mean square height. The labeled superscript letters within a row indicate a statistical difference in values between smooth Ti (A) and smooth + UVC Ti (a) groups, or between nano-Ti (B) and nano + UVC Ti (b) groups (*p* < 0.05).

## Data Availability

Data are available on request from the corresponding author.

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
