# Peer review of "Response of Human Gingival Fibroblasts and Porphyromonas gingivalis to UVC-Activated Titanium Surfaces"

_jfb, 2023, doi:10.3390/jfb14030137_

Round 1

Reviewer 1 Report

This study is very interesting and may have great clinical applicability. However, more studies must be carried out.

Materials and methods

“…silicon rubber burs, then cleaned ultrasonically…” - what brand of rubbers and what ultrasound machine were used (brand, city, country) are missing.

When the authors say: as previously report - they refer to something they have already reported, but this is not described in the text. These sentences are not correct. Authors must state who were the authors who used these preparations.

Between lines 94 and 136 - The authors do not refer to any guidelines. Are the recommended techniques their own?

Line 158 – Again, when the authors say: as previously report - they refer to something they have already reported, but this is not described in the text. These sentences are not correct. Authors must state who were the authors who used these preparations.

Until line 173 again, the authors do not refer to any guidelines.

Results

Table 2 – The authors used A and a, they must correct or explain why.

Error! Reference source not found” – what this mean??

Line 256 – authors should write which figure they are referring to.

In the analysis of the results, the texts that the authors write and that refer to the figures or tables found in the article must have the reference of the respective figure and/or table.

Discussion

Error! Reference source not found” – what this mean??

Author Response

Response to Reviewer 1 Comments

Manuscript ID: jfb-2132781

Title: Response of Human Gingival Fibroblasts and Porphyromonas Gingivalis to UVC-activated Titanium Surfaces

Authors: Yin Wen, Hao Dong, Jiating Lin, Xianxian Zhuang, Ruoting Xian, Ping Li*, Shaobing Li* 

We are truly grateful to you for sending us the reviewer’s comments on our manuscript. We have carefully revised the manuscript according to the reviewer’s comments. Suggestions from reviewers and editors enhance the richness of our content and its appeal to the readers, which are very helpful for our manuscript. During the revision period, we read the relevant references in detail and made the major revision of our manuscript as the chief editor and reviewers recommended. Our response to the reviewer’s comments is attached to this letter. The changes are highlighted in RED in the revised manuscript. We would also like to respond to the comments point-by-point as below.

Reviewer 1:

“…silicon rubber burs, then cleaned ultrasonically…” - what brand of rubbers and what ultrasound machine were used (brand, city, country) are missing.

[Reply] Thank you for your suggestion. We have modified the missing information. We also checked if there was any missing content in other sentences and tried to revise them based on this suggestion.

In line 93-96

…multiple sandpapers (Hermes-abrasives, Hamburg, Germany) up to P4000 and silicon rubber burs (0147, Qumo, Wuhan, China), then cleaned ultrasonically (KQ-700DE, Shumei, Kunshan, China) in acetone (GCRF, Guangzhou, China), ethanol (GCRF, Guangzhou, China), and ultrapure water (GOKU-B1, EWLL BIO, Guang-zhou, China)

When the authors say: as previously report - they refer to something they have already reported, but this is not described in the text. These sentences are not correct. Authors must state who were the authors who used these preparations.

[Reply] Thank you for your comments. We have revised the incorrect sentences accordingly. 

In line 97-98

The nano surfaces were obtained by anodization (labeled as N), as Wu et al. previously reported [23].

In line 101-104

The UVC pretreatment was performed on the smooth or nano Ti surfaces by using a UVC bactericidal lamp (TUV30W, Philips, Amsterdam, Netherland) with an intensity of 107 μW/cm2 (k = 253.7 nm) for 24 h at 25 °C, as Li et al. previously reported [24].

Between lines 94 and 136 - The authors do not refer to any guidelines. Are the recommended techniques their own?

[Reply] Thank you for this comment. All the experimental techniques recommended in our study followed the guidelines suggested by previous literature or the established protocols provided by the manufacturers of the laboratory reagents being used. We have modified the information to clarify the referred guidelines or protocols for each experiment.

In line 114-115

The physicochemical properties of the samples were characterized, as mentioned previously [25,26].

In line 125

Cell cultivation followed the protocols provided by Gibco®.

In line 132-133

Initial cell adhesion was evaluated by nucleus counting following the protocol provided by Meilun.

In line 139-140

The tetrazolium-based assay was used to assess the relative cell proliferation, as followed by Dojindo Laboratories.

In line 148-150

The morphology of cell spreading with the distribution of focal adhesions was visualized by immunofluorescence staining following the experimental protocol provided by Proteintech®[26].

Line 158 – Again, when the authors say: as previously report - they refer to something they have already reported, but this is not described in the text. These sentences are not correct. Authors must state who were the authors who used these preparations.

[Reply] Thank you for your suggestions. We have revised the incorrect sentences accordingly.

In line 189-190

as Di Giulio et al. previously described [28].

Until line 173 again, the authors do not refer to any guidelines.

[Reply] Thank you for this comment. We have modified the information to clarify the referred guidelines or protocols for each experiment.

In line 184

Microbial culture followed the protocols provided by AOBOX.

In line 192-193

The viability of P. g. was evaluated by live/dead bacterial staining following the protocols provided by AAT Bioquest.

In line 199-200

Biofilm formation was measured by crystal violet staining, as Kamble et al. previously reported [29].

In line 208-209

The adhesion of HGFs under P. g. pre-accumulation was explored by a co-culture model using the established protocol provided by Gao et al. [30].

Table 2 – The authors used A and a, they must correct or explain why.

[Reply] Thank you for this suggestion. We have corrected the labeled superscript letters and the footer of table 2 to explain the data more clearly.

In line 223-225

table 2

The labeled superscript letters within a row indicate a statistical difference in the values between Smooth Ti (A) and UVC-Smooth Ti (a) groups, or between Nano Ti (B) and UVC-nano Ti (b) groups (p < 0.05).

“Error! Reference source not found” – what this mean??

[Reply] Thank you for this comment. We are so sorry that the error caused by ENDNOTE occurred in the text. We have fixed the problems and inserted the reference again in the correct format.

Line 256 – authors should write which figure they are referring to.

[Reply] Thank you for this suggestion. We have inserted a citation of Figure 6I to the corresponding content.

In line 301-303

Biometric analysis showed that the cell area of HGFs at 2 h (Figure 6I) was significantly larger in the S+UVC group (p < 0.001) or N+UVC group (p = 0.002) compared with the untreated groups.

In the analysis of the results, the texts that the authors write and that refer to the figures or tables found in the article must have the reference of the respective figure and/or table.

[Reply] Thank you for this suggestion. We have carefully checked the missing information and added the references of the respective figures or tables to the text.

In line 275-279

After 0.5, 1, and 2 h of incubation, the S+UVC group (Figure 4A)… compared with the S group (p < 0.001 at 0.5 and 1 h; p = 0.018 at 2 h) (Figure 4C). In contrast, the N+UV group (Figure 4B) … compared with the N group at 0.5, 1, and 2 h, with significant differences… (Figure 4D).

In line 285-288

The relative cell proliferation of HGFs in the S+UVC group (Figure 5A)… no significant difference between N+UVC and N groups (Figure 5B)…

In order to make the description of Figure 6 and Figure 9 clearer, we have alphabetized each sub-image in these two figures and inserted the new references into the corresponding content.

Figure 6

In line 296-304

HGFs in the S+UVC group (Figure 6B) extensively stretched… In contrast, the S group (Figure 6A) … Similarly, HGFs in the N+UVC group (Figure 6D) extended better than in the N group (Figure 6C) … Biometric analysis showed that the cell area of HGFs at 2 h (Figure 6I) was significantly larger… After 24 h of incubation, HGFs on all the surfaces (Figure 6E, F, G and H) were thoroughly elongated…

In line 312-314

Figure 6. …(A) smooth Ti, (B) smooth+UVC Ti, (C) nano Ti, (D) nano+UVC Ti and at 24 h for (E) smooth Ti, (F) smooth+UVC Ti, (G) nano Ti, (H) nano+UVC Ti; Biometric analysis of (I) cellular area, (J) perimeter, and (K) Feret’s diameter at 2 h.

Figure 9

In line 347-351

The S+UVC group (Figure 9B and F) had more extensive cell spreading and less bacterial accumulation compared with the S group (Figure 9A and E) while the N+UVC group (Figure 9D and H) failed to show a significant change in cell spreading com-pared with the N group (Figure 9C and G), although the P. g. pre-accumulation declined.

In line 355-357

… images in magnification 2000 × for (A) smooth Ti, (B) smooth+UVC Ti, (C) nano Ti, (D) nano+UVC Ti and in magnification 1, 000 × for (E) smooth Ti, (F) smooth+UVC Ti, (G) nano Ti, (H) nano+UVC Ti.

“Error! Reference source not found” – what this mean??

[Reply] Thank you very much for the suggestion. We are so sorry about the error. We have solved the problems and inserted the reference again in the correct format.

Reviewer 2 Report

Comments for Authors:

The present work on "Response of Human Gingival Fibroblasts and Porphyromonas Gingivalis to UVC-activated Titanium Surfaces" is interesting but there are several important points authors need to address before its consideration. 

1.  Introduction is not focussed on the material's point of view at all. 

2.  Why Ti is widely used must be explained properly. 

3. What is the novelty of this study? UV irradiation has been done previously by many researchers. 

4. What changes to the materials are done must be explored more. 

5. The surface roughness profile must be incorporated. 

6. EDS study should be done to confirm the elemental presence. 

7. How the nanotube was confirmed? 

8. Many error formats must be removed. e.g., Error! Reference source not found.G, etc.

9. Briefly explain "2−ΔΔCT method".

10. Expand each abbreviation properly. 

11. What are the reasons to get "static WCA at 0°" after UVC?

12. The study itself has many limitations. 

Author Response

Response to Reviewer 2 Comments

Manuscript ID: jfb-2132781

Title: Response of Human Gingival Fibroblasts and Porphyromonas Gingivalis to UVC-activated Titanium Surfaces

Authors: Yin Wen, Hao Dong, Jiating Lin, Xianxian Zhuang, Ruoting Xian, Ping Li*, Shaobing Li*

We are truly grateful to you for sending us the reviewer’s comments on our manuscript. We have carefully revised the manuscript according to the reviewer’s comments. Suggestions from reviewers and editors enhance the richness of our content and its appeal to the readers, which are very helpful for our manuscript. During the revision period, we read the relevant references in detail and made the major revision of our manuscript as the chief editor and reviewers recommended. Our response to the reviewer’s comments is attached to this letter. The changes are highlighted in RED in the revised manuscript. We would also like to respond to the comments point-by-point as below.

Reviewer 2:

  1. Introduction is not focussed on the material's point of view at all.

[Reply] Thank you for this valuable suggestion. We reorganized the ideas and improved the writing structures in the second paragraph of the Introduction to make it focus on the material’s point of view as much as possible.  

In line 39-60

Various surface modification strategies have been applied to improve soft tissue integration, including machining, acid-etching, argon plasma, laser melting, and ano-dization [7]. However, most methods are insufficient to inhibit bacterial accumulation because either cells or microbes can be affected by the surface properties of the sub-strates [8,9]. From the aspect of cells, the anodized nano Ti surface was observed bet-ter adhesion of gingival fibroblasts compared with other types of surfaces. Thereby, the anodization has been considered as a promising technique for surface modifica-tion of Ti materials [10,11]. On the other hand, the machined smooth Ti surface dis-played less biofilm formation and more convenient plaque control than other surface designs. The surface is characterized as an arithmetic mean roughness value of fewer than 0.2 μm, which is currently recommended for abutments [12,13]. Thus, it would be challenging to fabricate a Ti abutment surface that synergistically improves cell ad-hesion and inhibits bacterial attachment.

  1. Why Ti is widely used must be explained properly.

[Reply] Thank you for this comment. The reason why Titanium (Ti) has been widely used may relate to its excellent performance in biocompatibility, corrosion resistance and mechanical properties. We briefly introduced this reason in the first paragraph of the Introduction.

In line 30-31

Titanium (Ti) has been the preferred choice for the fabrication of dental implants due to its excellence in biocompatibility, corrosion resistance and mechanical properties [1,2].

To improve coherence and cohesion, we also revised the following sentences.

In line 32-36

In the Ti dental implant system, a stable transmucosal region of the abutments and necks, or soft tissue integration, plays a critical role in achieving long-term success in dental implant restorations [3,4]. Unfortunately, the peri-implant tissue is susceptible to bacterial invasion compared with the periodontal tissues of natural teeth and the bioinert of Ti materials may aggravate this adverse situation [5,6].

  1. What is the novelty of this study? UV irradiation has been done previously by many researchers.

[Reply] Thank you for this comment.

Our study was the first to investigate the effect of UVC on two types of surface design that are currently applied in commercial abutments, and to compare the in vitro response of HGFs and P. g. not only by monoculture methods but also by co-culture of the single cells and microbes.

In fact, the effect of UV photofunctionalization may depend on the parameters including the wavelength of UV light, the duration of UV irradiation, the types of materials, the surface properties of the materials, and even the type of organisms. While there have been many pieces of research focused on UV irradiation, the effect of UV on Ti with different surface structures especially the nanostructures are rarely studied. Moreover, these studies paid little attention to the clinical problems of gingival healing and the accumulation of periodontal pathogens. From this perspective, our study was the first to bridge the gap between UVC photofunctionalization and soft tissue integration from the view of clinical practice. Our study aimed to optimize the surface design of Ti implant abutments which might potentially be a more effective solution for improving the long-term stability of Ti dental implants.

We emphasized the novelty of this study in the last two paragraphs of the Introduction.

In line 61-78

Ultraviolet (UV) photofunctionalization is expected to overcome this issue. It refers to a phenomenon of modification of Ti surfaces that occurs after UV treatment, including the alteration of physicochemical properties and the improvement of bioactivity [14]. Current studies revealed that UV-activated Ti implant surfaces enhanced bone regeneration in vitro and in vivo [15,16]. Meanwhile, UV irradiation, which rarely changes the structure of the objects, has been widely used in surface sterilization for centuries, and it has become attractive for its potential application in light-based antimicrobial therapies [17-19]. Interestingly, the effect of UV photofunctionalization may depend on varying factors including the wavelength of UV light, the duration of UV irradiation, the types of materials, the surface properties of the materials, and even the type of organisms. For example, it was reported that Ultraviolet C (UVC; 100-280 nm) pretreatment generated more hydrophilic groups and achieved a higher level of cell adhesion than Ultraviolet A (UVA; 315-400 nm) [20]; the hydrophilic groups increased with a longer duration of UV irradiation [21]; UV pretreatment promoted osteoblastic differentiation of cells on Ti surface with anatase-enriched coating [22]. While there have been many pieces of research focused on UV irradiation, the effect of UV on Ti with different surface structures especially the nanostructures are rarely studied. Moreover, these studies paid little attention to the clinical problems of gingi-val healing and the accumulation of periodontal pathogens.

In line 83-84

…different Ti-based surfaces, which corresponded to two representative types of surface design that are currently applied in commercial abutments.

  1. What changes to the materials are done must be explored more.

[Reply] Thank you for this suggestion. We further explored the changes in the materials after the surface modification in the section of Result and Discussion. Based on different preparations, the smooth Ti and nano Ti were varied in surface morphology and roughness. After UVC pretreatment, both two surfaces observed a significant alteration of the surface wettability and chemical composition while little changes in the surface structure.

Revision of the Result:

In line 226-227

The surface roughness of the smooth-based Ti was no more than 20 nm in Sa while the nano Ti was slightly rougher which turned out to be 20-30 nm in Sa (Table 2).

In line 243-245

Therefore, the smooth Ti surfaces were hydrophobic while the nano Ti surfaces were hydrophilic. After UVC pretreatment, both surfaces acquired superhydrophilicity.

Revision of the Discussion:

In line 360-382

UV photofunctionalization has been reported to improve cell adhesion and reduce biofilm formation on Ti-based surfaces, which may potentially be favorable for soft tissue integration of Ti dental implants. In this study, smooth Ti and nano Ti surfaces were chosen as two representative surface designs that are currently used in commercial abutments. Compared with the smooth-based Ti surfaces (Figures 1A and C) that exhibited scratch-like patterns on a micro scale, the nano-based Ti surfaces (Figures 1B and D) displayed a wide range of TiO2 nanotubes with increased surface roughness (Figures 1E and F). It was previously evidenced that the introduction of nano texture augmented the contact area and increased the surface wettability, which had a higher affinity to cells [32]. However, such multi-structured topography could be less effective for inhibiting bacterial accumulation and removing bacterial debris than the smooth surface [13]. For this reason, UVC pretreatment was applied in this study to optimize the bioactivity and antimicrobial activity of different Ti surfaces. After UVC pretreatment, both smooth and nano Ti surfaces got a static WCA at 0° (Table 2) as evidence of super-hydrophilic transition (Figure 1 G and H) without significant changes in surface topography or roughness, which was consistent with the previous research [33]. This phenomenon was explained by the removal of carbohydrate contaminants and the production of hydrophilic chemicals [25]. It was also confirmed in our study by the result of EDS (Figure 2) and XPS (Figure 3), which exhibited increased hydroxyl groups (Figure 3A and B) and decreased carbon elements on the UVC-pretreated surfaces compared with the untreated ones. The related mechanism of the alteration in chemical properties may differ in the wavelengths of UV light. It was reported that UVA-induced TiO2 photocatalytic reactions while UVC generated hydrophilic groups by directly breaking the chemical bonds rather than photocatalysis [34].

5. The surface roughness profile must be incorporated.

[Reply] Thank you for this suggestion. We added the surface roughness profile to Figure 1 and modified the figure caption accordingly.

In line 250-251

Figure 1. AFM images with the surface roughness profile at a range of 5 μm × 5 μm for (E) smooth Ti and (F) nano Ti before and after UVC irradiation;

Figure 1

  1. EDS study should be done to confirm the elemental presence.

[Reply] Thank you for this suggestion. We supplemented the result of EDS analysis with another figure for this study, which demonstrated a similar outcome for the elemental presence as the XPS analysis. Data was updated in table 2 and the details were described in the Materials & Methods and Results.

In line 118-120

The surface chemistry of the samples was investigated by energy-dispersive X-ray spectroscopy (EDS, XFlash 6|30, Bruker, Karlsruhe, Germany) and X-ray photoelectron spectroscopy (XPS; K-Alpha, Thermo Fisher Scientific, Waltham, USA).

In line 254-258

The main chemical compositions of the samples were all related to the O, C, and Ti elements which were measured by EDS and XPS analysis (Table 2). EDS showed that both the S+UVC group (Figure 2A) and the N+UVC group (Figure 2B) exhibited an increased content of oxygen and a decreased content of carbon compared with the control groups. This result was confirmed by XPS.

In line 222

Table 2. Physiochemical properties of the samples.

table 2

In line 223-225 (footer of table 2)

The labeled superscript letters within a row indicate a statistical difference in values between smooth Ti (A) and smooth+UVC Ti (a) groups, or between nano Ti (B) and nano+UVC Ti (b) groups (p < 0.05).

In line 269 (caption of figure 2)

Figure 2. Element peaks of EDS analysis for (A) smooth Ti and (B) nano Ti.

Figure 2

  1. How the nanotube was confirmed?

[Reply] Thank you for this comment. According to previous studies, the TiO2 nanotube is typically characterized by tube-like structures in diameter 10–300 nm, which can be confirmed by SEM [1]. Under high magnification (50, 000 ×) of SEM in our study, the anodized Ti surface displayed a wide range of tubular structures in tight and orderly arrangement. Each unit of the structures was approximately 90 nm in diameter like empty nano-scale test tubes opened at the top and closed at the bottom, suggesting the existence of TiO2 nanotube arrays.

In line 232-235

Under high magnification, it displayed a wide range of tubular structures in tight and orderly arrangement (Figure 1D). Each unit of the structures was approximately 90 nm in diameter (Table 2) like empty nano-scale test tubes opened at the top and closed at the bottom, suggesting the existence of TiO2 nanotube arrays.

  1. Many error formats must be removed. e.g., Error! Reference source not found.G, etc.

[Reply] Thank you for this suggestion. We are so sorry for these errors. We have fixed the problems and removed the wrong citation. The missing references have been inserted again in the correct format.

  1. Briefly explain "2−ΔΔCTmethod".

[Reply] Thank you for this suggestion. Quantifying the relative changes in gene expression using real-time PCR requires certain equations, assumptions, and the testing of these assumptions to properly analyze the data. And the 2−ΔΔCT method is a convenient way to calculate the relative changes in gene expression determined from real-time quantitative PCR experiments.

In line 175-176

The relative changes in mRNA expression determined from qPCR experiments were calculated by the 2−ΔΔCT method.

  1. Expand each abbreviation properly.

[Reply] Thank you for this comment. We checked the full text carefully to find any unexpanded abbreviations in the first place and revised them. 

In line 163-164

the Minimum Information for Publication of Quantitative Real-Time PCR Experiments (MIQE)

In line 172-173

Quantitative polymerase chain reaction (qPCR)

In line 176-177

Glyceraldehyde-3-phosphate dehydrogenase (GAPDH)

  1. What are the reasons to get "static WCA at 0°" after UVC?

[Reply] Thank you for this comment. This phenomenon was explained by the removal of carbohydrate contaminants and the production of hydrophilic chemicals induced by UVC irradiation. As the increased accumulation of hydrophilic groups, the water droplet spread more extensively on the surfaces. We further explained the reasons in the first paragraph of the Discussion.

In line 375-382

This phenomenon was explained by the removal of carbohydrate contaminants and the production of hydrophilic chemicals [25]. It was also confirmed in our study by the result of EDS (Figure 2) and XPS (Figure 3), which exhibited increased hydroxyl groups (Figure 3A and B) and decreased carbon elements on the UVC-pretreated surfaces compared with the untreated ones. The related mechanism of the alteration in chemical properties may differ in the wavelengths of UV light. It was reported that UVA induced TiO2 photocatalytic reactions while UVC generated hydrophilic groups by directly breaking the chemical bonds rather than photocatalysis [34].

  1. The study itself has many limitations.

[Reply] Thank you very much for this suggestion. The major limitations in this study are as followed. Firstly, the explanation for the inhibited adhesive behavior of HGFs on UVC-pretreated nano Ti has not yet been confirmed. Secondly, the molecular mechanisms of changes in the function of HGFs should be further explored. Thirdly, it remains unknown whether the long-term effects of UVC pretreatment would be advantageous or disadvantageous for soft tissue sealing since all these in vitro models were only observed in short periods. In addition, the effect exerted on a flat disc sample might be significantly different from that on the curved surface of the abutment components. Moreover, the present study lacks in vivo studies.

In line 456-474

Nevertheless, the biological response to the implant materials was extremely complicated, and it is still poorly understood for the optimal design of dental implant abutments. Our findings can provide new insights into the potential application of UVC pretreatment on Ti surfaces for enhancing soft tissue integration and preventing peri-implant diseases. The major limitations of this study included as follows. Firstly, the mechanism for the inhibited adhesive behavior of HGFs on UVC-pretreated nano Ti has been not confirmed yet. Secondly, the molecular mechanisms of changes in the function of HGFs should be further explored. Thirdly, it remains unknown whether the long-term effects of UVC pretreatment exhibit the pros and cons of soft tissue seal-ing because most in vitro models were observed in short periods. In addition, the effect exerted on a flat disc sample might be significantly different from that on the curved abutment surface. More importantly, the present study lacks in vivo verification.

References

Chopra, D.; Gulati, K.; Ivanovski, S. Understanding and optimizing the antibacterial functions of anodized nano-engineered titanium implants. Acta biomaterialia 2021, 127, 80-101, doi:10.1016/j.actbio.2021.03.027.

Reviewer 3 Report

The article ‘Response of human gingival fibroblasts and Porphyromonas Gingivalis to UVC-activated titanium surfaces’ by Yin Wen, Hao Dong, Jiating Lin, Xianxian Zhuang, Ruoting Xian, Ping Li and Shaobing Li is an very interesting manuscript. The authors put a lot of work in research and preparing of manuscript. The work describes the effect of UVC pretreatment on the response of human gingival fibroblasts (HGFs) and Porphyromonas Gingivalis to Ti-based implant surfaces. This article fits the subject of the Journal of Functional Biomaterials. The manuscript may be published after a slight correction. Some minor flaws are listed below:

Line 97,98 – XPS is X-ray Photoelectron Spectroscopy not microscopy

References:

- text ‘Error! Reference source not found’ should be removed.

- reference ‘Komine et al.” has no number.

The article ‘Response of human gingival fibroblasts and Porphyromonas Gingivalis to UVC-activated titanium surfaces’ by Yin Wen, Hao Dong, Jiating Lin, Xianxian Zhuang, Ruoting Xian, Ping Li and Shaobing Li is an very interesting manuscript. The authors put a lot of work in research and preparing of manuscript. The work describes the effect of UVC pretreatment on the response of human gingival fibroblasts (HGFs) and Porphyromonas Gingivalis to Ti-based implant surfaces. This article fits the subject of the Journal of Functional Biomaterials. The manuscript may be published after a slight correction. Some minor flaws are listed below:

Line 97,98 – XPS is X-ray Photoelectron Spectroscopy not microscopy

References:

- text ‘Error! Reference source not found’ should be removed.

- reference ‘Komine et al.” has no number.

Author Response

Response to Reviewer 3 Comments

Manuscript ID: jfb-2132781

Title: Response of Human Gingival Fibroblasts and Porphyromonas Gingivalis to UVC-activated Titanium Surfaces

Authors: Yin Wen, Hao Dong, Jiating Lin, Xianxian Zhuang, Ruoting Xian, Ping Li*, Shaobing Li*

We are truly grateful to you for sending us the reviewer’s comments on our manuscript. We have carefully revised the manuscript according to the reviewer’s comments. Suggestions from reviewers and editors enhance the richness of our content and its appeal to the readers, which are very helpful for our manuscript. During the revision period, we read the relevant references in detail and made the major revision of our manuscript as the chief editor and reviewers recommended. Our response to the reviewer’s comments is attached to this letter. The changes are highlighted in RED in the revised manuscript. We would also like to respond to the comments point-by-point as below.

Reviewer 3:

Line 97,98 – XPS is X-ray Photoelectron Spectroscopy not microscopy

[Reply] Thank you for this suggestion. We have corrected the wrong information accordingly.  

In line 119-120

X-ray photoelectron spectroscopy

- text ‘Error! Reference source not found’ should be removed.

[Reply] Thank you for this comment. We are so sorry for these errors. We have fixed the problems and removed the wrong citation. The missing references have been inserted again in the correct format.

- reference ‘Komine et al.” has no number.

[Reply] Thank you very much for this comment. We have modified the missing and incorrect information accordingly.

Round 2

Reviewer 2 Report

The present work on "Response of Human Gingival Fibroblasts and Porphyromonas 2 Gingivalis to UVC-activated Titanium Surfaces" is interesting. But, minor corrections need to be made before acceptance.
1. All the questions asked to the authors have been answered properly.
2. The revised manuscript (jfb-2132781-peer-review-v2) has been significantly improved.
3. The roughness parameters Sa and Sq should be defined in the text.
4. The font size of the text and values in Figures 1E to 1H should be increased up to a visible scale.
5. Angle should be mentioned in the Figures 1G and 1H.

Author Response

Response to Reviewer 2 Comments - Round 2

Manuscript ID: jfb-2132781

Title: Response of Human Gingival Fibroblasts and Porphyromonas Gingivalis to UVC-activated Titanium Surfaces

Authors: Yin Wen, Hao Dong, Jiating Lin, Xianxian Zhuang, Ruoting Xian, Ping Li*, Shaobing Li*

Thank you very much for sending us the reviewer’s second-round comments on our manuscript. We have carefully revised the manuscript according to the reviewer’s comments. Our response to the reviewer’s comments is attached to this letter. The changes are highlighted in RED in the revised manuscript. We would also like to respond to the comments point-by-point as below.

Reviewer 2:

  1. The roughness parameters Sa and Sq should be defined in the text.

[Reply] Thank you for this suggestion. We have added the definition of Sa and Sq in the footer of table 2.

In line 208

Sa: arithmetic mean height; Sq: root mean square height.

  1. The font size of the text and values in Figures 1E to 1H should be increased up to a visible scale.

[Reply] Thank you for this suggestion. We have revised the pictures accordingly.

  1. Angle should be mentioned in the Figures 1G and 1H.

[Reply] Thanks for this comment. We have added this information to the figures of WCA.
